# A Dynamic Topology Optimization Method for Tactical Edge Networks Based on Virtual Backbone Networks

**DOI:** 10.3390/s24175489

**Published:** 2024-08-24

**Authors:** Zhixiang Kong, Zilong Jin, Chengsheng Pan

**Affiliations:** 1School of Automation, Nanjing University of Science and Technology, Nanjing 210094, China; kxc5072@163.com; 2School of Software, Nanjing University of Information Science and Technology, Nanjing 210044, China; zljin85@163.com; 3School of Electronics and Information Engineering, Nanjing University of Information Science and Technology, Nanjing 210044, China

**Keywords:** tactical edge networks, connected dominating sets, topology optimization, marine predator algorithm

## Abstract

To address the issues of low survivability and communication efficiency in wireless sensor networks caused by frequent node movement or damage in highly dynamic and high-mobility battlefield environments, we propose a dynamic topology optimization method based on a virtual backbone network. This method involves two phases: topology reconstruction and topology maintenance, determined by a network coverage threshold. When the coverage falls below the threshold, a virtual backbone network is established using a connected dominating set (CDS) and non-backbone node optimization strategies to reconstruct the network topology, quickly restore network connectivity, effectively improve network coverage, and optimize the network structure. When the coverage is above the threshold, a multi-CDS scheduling algorithm and slight position adjustments of non-backbone nodes are employed to maintain the network topology, further enhancing network coverage with minimal node movement. Simulations demonstrate that this method can improve coverage and optimize network structure under different scales of network failures. Under three large-scale failure operational scenarios where the network coverage threshold was set to 80%, the coverage was enhanced by 26.12%, 15.88%, and 13.36%, and in small-scale failures, the coverage was enhanced by 7.55%, 4.90% and 7.84%.

## 1. Introduction

In recent years, the U.S. military has proposed new combat styles such as multi-domain joint operations [1], mosaic warfare [2], distributed operations [3], and swarm drone operations [4]. These combat styles emphasize dispersed forces and intelligent collaboration. A typical example is the tactical edge network [5], whose structural composition is shown in Figure 1. This network integrates combat soldiers with unmanned combat units such as drones, ground robots, and unmanned combat vehicles, forming small, sophisticated, and powerful joint units. These units are capable of rapid and precise strikes, thereby creating advantageous windows across the entire battlefield and accomplishing combat missions such as key area seizure and urban warfare.

The tactical edge network, designed specifically to meet the communication needs of modern battlefields and complex environments, features unique structures and functionalities to ensure stable operation under extreme conditions. This network system deeply integrates the technological characteristics of wireless sensor networks (WSNs) [6] and mobile ad hoc networks (MANETs) [7], embodying heterogeneity, high dynamism, large-scale deployment, self-organization, and self-healing capabilities. In complex battlefield scenarios, the tactical edge network plays a crucial role in reconnaissance and surveillance, target tracking, environmental monitoring, information collection, and communication.

However, during the operation of the tactical edge network, nodes are susceptible to enemy attacks, leading to frequent node mobility and resulting in changes in the network’s center of gravity. In such a highly dynamic and topologically variable battlefield environment, the network may face issues like node failures, causing a decline in network performance or even large-scale breakdowns. Therefore, it is necessary to design a topology optimization method, which can dynamically adjust the network structure to improve coverage, reduce latency, and enhance the network’s robustness and reliability.

Existing network topology optimization methods can be divided into three categories: power control-based, clustering-based, and connected dominating set-based methods.

Most power control methods are based on direction, node degree, and proximity graphs. The representative algorithm based on direction is cone-based topology control (CBTC) [8], which ensures that there is at least one adjacent node within the effective area by dynamically adjusting the transmission angle. The representative algorithm based on node degree is the local mean algorithm (LMA) [9], which dynamically adjusts transmission power to fall within a specified range by constraining the upper and lower limits of node degree, achieving adaptive power control. Proximity graph-based methods abstract the network as vertices and edges, using different graph constructions to implement power control. Representative algorithms include the unit disk graph (UDG) [10], Gabriel graph (GG) [11], relative neighborhood graph (RNG) [12], and Delaunay triangulation graph (DTG) [13]. These algorithms use different constraints and rules to define vertex–edge relationships to achieve power control. Current power control algorithms either require precise node location information or involve large amounts of information exchange between neighboring nodes and do not adequately consider the network’s robustness.

Clustering-based methods periodically elect some nodes to act as cluster heads, subdividing the network into regions managed by respective cluster heads for data aggregation and forwarding. BM et al. [14] proposed a topology optimization method for heterogeneous wireless sensor networks based on energy-saving clustering based on particle swarm optimization is proposed, which effectively improves the life cycle through bionics methods. This strategy also adopts a meticulously designed node active/sleep scheduling mechanism to arrange work and rest cycles efficiently, reducing unnecessary energy consumption. A. Ali et al. [15] proposed an optimization-based clustering algorithm for vehicular ad hoc networks, the random operator of the algorithm and the proper maintenance of the balance state between exploration and exploitation operations enable the proposed algorithm to escape from local optimality and provide a global optimal solution. Although clustering-based methods can simplify network management in some cases, their heavy burden on cluster head nodes, complex elections, and high maintenance costs limit their application in dynamic and large-scale networks.

The connected dominating set (CDS) method ensures the global connectivity of the entire network by selecting a node set so that each node is connected to at least one dominating node. This can effectively reduce the number of transmission hops and delays and improve the efficiency of data transmission. The CDS method can adapt to dynamically changing network topology. For example, when nodes or links in the network change, the network structure can be quickly adjusted by recalculating the dominating set to ensure the stability and reliability of the network. Rizvi et al. [16] developed the energy-efficient CDS (EECDS) algorithm, which leverages the CDS principle to establish a backbone transmission channel, ensuring efficient communication between nodes. Feng He et al. [17] proposed a centralized approximation algorithm for CDS reconstruction to solve the minimum fault-tolerant CDS problem in a given disk graph with bidirectional links (DGB) and significantly improved network performance. Fu D. et al. [18] proposed an algorithm based on a greedy strategy to construct minimum connected dominating set (MCDS). The greedy algorithm selects nodes step by step, so that each step can maximize the number of covered nodes that are not currently covered and ensure that the selected node set is always connected. Mohanty et al. [19] proposed a novel concept, the pseudo-dominant set, to assist in constructing a CDS. The pseudo-dominant set does not require each node to be directly connected to a dominating node like a traditional dominating set but allows some nodes to be indirectly connected to the dominating node through other nodes, thereby reducing the restrictions on selecting nodes. Wang et al. [20] proposed a method to construct a virtual backbone network (VBN) based on the graph theory d-hop CDS (d-CDS), which can reduce the excessive routing overhead while ensuring the connectivity of multi-hop links and achieve a better balance between response time and maintenance cost.

Despite the maturity of MCDS construction algorithms in WSNs and MANETs, frequent topology changes in tactical edge networks present challenges. Improving MCDS algorithms for the dynamic nature of tactical edge networks is thus a key research direction in topology optimization. Based on this, we propose a dynamic topology optimization framework based on a virtual backbone network, as shown in Figure 2. For a given target network monitoring area, environmental information of the monitoring area is inputted. According to the current network coverage situation, the topology optimization of the tactical edge network is divided into two different scenarios: large-scale failure topology reconstruction and small-scale topology maintenance. The dynamic topology optimization is carried out in these two phases.

The main research of this paper includes the following:

1. In the topology reconstruction phase, backbone nodes are first selected based on the CDS algorithm to construct a virtual backbone network. Then, with the current network coverage rate and the average distance from non-backbone nodes to backbone nodes as optimization goals, and the current network algebraic connectivity as a constraint, the positions of non-backbone nodes are iteratively optimized using an enhanced multi-objective marine predator algorithm. This quickly improves network robustness while further optimizing communication and coverage capabilities, ensuring rapid network recovery and adaptive interconnection of critical nodes.

2. In the topology maintenance phase, backbone network topology is updated according to the maximum similarity principle using a multi-CDS scheduling backbone maintenance algorithm in the current time slot. Then, with the current network coverage rate and node movement distance as optimization goals and the current network algebraic connectivity as a constraint, the positions of non-backbone nodes are fine-tuned using an enhanced multi-objective marine predator algorithm to further improve network coverage while minimizing energy consumption, constructing the optimal network topology for the current time slot.

3. Applying this scheme to large-scale and small-scale network failures, simulation experiments show that in three different scenarios, the network can achieve rapid interconnection, improve coverage, and construct optimal topologies whether facing large- or small-scale failures.

## 2. System Model

### 2.1. Network Topology Model

Assuming that the nodes in the tactical edge network [21] are homogeneous, i.e., each node has the same communication radius, the topology of the tactical edge network can be abstracted as an undirected connected graph, represented by G(V,E). In this representation, the network nodes are denoted by the vertex set $V=\{v_{1},v_{2},v_{3},\ldots ,v_{n}\}$, and the links are represented by the edge set $E=\{e_{1},e_{2},e_{3},\ldots ,e_{m}\}$. For $\forall e_{i}\in E$, there exists a corresponding $(v_{p},v_{q})\in V$ with $e_{i}$. Considering the high topological variability of the tactical edge network, the network operation time is divided into several time slots. Thus, the network topology at time $t_{k}$ is represented as $G(V^{t_{k}},E^{t_{k}})$.

In this paper, the nodes in the tactical edge network are equipped with omnidirectional antennas, meaning that any node within its communication radius can receive the signal. In a real complex battlefield environment, the ability of nodes to communicate normally depends on many factors, such as the remaining energy of the nodes and the current combat environment. However, this paper focuses on the problem of network topology construction and maintenance under high topological variability. To simplify the process of establishing the network topology, we mainly consider the distance between nodes as the only condition for establishing communication between two nodes. Therefore, let $R_{c}$ be the communication radius between nodes. If nodes $v_{p}$ and $v_{q}$ satisfy $d\left(v_{i},v_{j}\right)\leq R_{c}$, it means that the two nodes can communicate with each other and nodes $v_{p}$ and $v_{q}$ are neighbor nodes, i.e., $v_{p}\in N_{q}$ and $v_{q}\in N_{p}$, where $N_{p}$ and $N_{q}$ represent the one-hop neighbor node sets of nodes $v_{p}$ and $v_{q}$, respectively. Based on this, this paper provides the following relevant definitions:

**Definition** **1.**
*Node Degree $D(v_{p})$: For any node $v_{p}$, the total number of neighbor nodes within one hop is called the degree of the node.*


**Definition** **2.**
*Edge Weight $w(e_{<v_{p},v_{q}>})$: For any edge <$v_{p},v_{q}$>, the sum of the degrees of the two connected nodes $v_{p}$ and $v_{q}$ is taken as the negative value, which is the weight of the connecting edge.*


**Definition** **3.**
*Node Weight $w(v_{p})$: For any node, the node weight is set based on its current node degree and the degree of its neighbor nodes. The specific calculation is as follows:*

(1)
$$w\left(v_{p}\right)=\sum_{v_{q}\in N_{p}}\frac{w\left(v_{q}\right)*w\left(e_{<v_{p},v_{q}>}\right)}{D(v_{p})}$$



**Definition** **4.**
*CDS $D(v_{p})$: All CDS sets generated in the undirected graph.*


**Definition** **5.**
*Minimum CDS $CDS_{k}$: The minimum CDS selected at the current time, where <$v_{p},v_{q}$> represents the node set in the CDS, i.e., the backbone node set, and $v_{p}$ represents the link set in the CDS, i.e., the link set between backbone nodes.*


**Definition** **6.**
*CDS Weight $w(CDS_{k})$: For any CDS, the sum of the weights of all dominating nodes is the weight of the CDS, specifically calculated as follows:*

(2)
$$w\left(CDS_{k}\right)=\sum_{v_{p}\in CDS_{k}}w(v_{p})$$



**Definition** **7.**
*$Cost_{t_{k}}$ represents the total update cost of the network at the current time. Considering that the network nodes in this experiment are homogeneous, this paper assumes that updating each node requires a certain communication cost. Therefore, the total update cost is determined by the number of nodes updated during the final maintenance phase. This cost can be simplified as the number of updated nodes, calculated as follows:*

(3)
$$Cost_{t_{k}}=\sum_{i=1}^{N}C(v_{i})$$


(4)
$$C\left(v_{i}\right)=\left\{\begin{array}{cc} 1, & \mathrm{if}\;v_{i}\;\mathrm{will}\;\mathrm{be}\;\mathrm{updated}\\ 0, & \mathrm{if}\;v_{i}\;\mathrm{will}\;\mathrm{not}\;\mathrm{be}\;\mathrm{updated} \end{array}\right.$$

*where N is the total number of network nodes, and $C(v_{i})$ represents whether node $v_{i}$ will be updated. If updated, $C(v_{i})=1$; otherwise, $C(v_{i})=0$.*


### 2.2. Network Coverage Model

Figure 3 is a classic schematic diagram of network coverage model. Assuming the tactical edge network monitoring area is a two-dimensional plane $S=l_{1}\times l_{2}$, the network monitoring area is discretely divided into $l_{1}\times l_{2}$ monitoring point grids. The network monitoring points can be defined as $M\;=\{m_{1},\;m_{2},\ldots ,\;m_{j},\ldots ,{m_{l_{1}}}_{\times l_{2}}\}$, where the coordinates of monitoring point $m_{j}$ are $\left\{x_{j},y_{j}\right\}$, $i\in (1,l_{1}\times l_{2})$. The geometric center of each monitoring point is the target location for coverage optimization.

Assuming there are *n* homogeneous sensor nodes, the set of nodes is $Z=\{z_{1},z_{2},\ldots ,z_{i},\ldots ,z_{n}\}$, where the coordinates of node $z_{i}$ are $\left\{x_{i},y_{i}\right\}$, $i\in \left(1,n\right)$. Each node has a sensing radius $R_{s}$ and a communication radius $R_{c}$, with $R_{c}=2R_{s}$ to ensure the connectivity of the tactical edge network. The distance between node $z_{i}$ and monitoring point $m_{j}$ is defined as follows:(5)$$d\left(z_{i},m_{j}\right)=\sqrt{{\left(x_{i}-x_{j}\right)}^{2}+{\left(y_{i}-y_{j}\right)}^{2}}$$

The probability that monitoring point $m_{j}$ is sensed by node $z_{i}$ is modeled using a Boolean sensing model. When the distance between monitoring point $m_{j}$ and node $z_{i}$ is not greater than the sensing radius, the monitoring point grid is considered covered. The specific formula is as follows:(6)$$p_{\mathrm{cov}}\left(z_{i},m_{j}\right)=\left\{\begin{array}{cc} 1, & if\;d\left(z_{i},m_{j}\right)\leq R_{s}\\ 0, & else\; \end{array}\right.$$

In the monitoring area, each monitoring point can be sensed by multiple nodes simultaneously. The combined sensing probability $C_{p}$ of all nodes for monitoring point $m_{j}$ is defined as follows:(7)$$C_{p}\left(Z,m_{j}\right)=1-\prod_{i=1}^{n}\left[1-p_{\mathrm{cov}}\left(z_{i},m_{j}\right)\right]$$

From the paragraph, the total coverage rate $R_{\mathrm{cov}}$ of the area can be calculated as follows:(8)$$R_{\mathrm{cov}}=\frac{\sum_{j=1}^{l_{1}\times l_{2}}C_{p}\left(Z,m_{j}\right)}{l_{1}\times l_{2}}$$

### 2.3. Location Optimization Model

In practical battlefield scenarios, there is often a demand for multi-objective optimization. To achieve rapid network recovery and further optimize the topology, after constructing the backbone network, the next time slot positions of nodes can be adjusted. Based on this, this paper proposes a non-backbone node position optimization strategy using a multi-objective enhanced marine predator algorithm. Through mathematical modeling, the problem is transformed into a constrained dual-objective optimization problem, can be calculated as follows:(9)$$\begin{array}{c} opt.\left\{\begin{array}{c} \max\sum_{v_{i}\in V}\sum_{v_{j}\in V}R_{e_{ij}}\left[t\right]\\ \max R_{\mathrm{cov}}=\frac{\sum_{j=1}^{l_{1}\times l_{2}}C_{p}\left(Z,m_{j}\right)}{l_{1}\times l_{2}} \end{array}\right.\\ s.t.\;\lambda_{2}\left(L\right)>0 \end{array}$$
where algebraic connectivity $\lambda_{2}\left(L\right)$ is chosen as the constraint, representing the second smallest eigenvalue of the network Laplacian matrix. Algebraic connectivity measures the connectivity and robustness of the network; $\lambda_{2}\left(L\right)$ ensures that the network is connected, guaranteeing resilience against destruction.
(10)$$R_{e_{i,j}}\left[t\right]=B\log\left(1+\frac{p\left[t\right]\eta_{0}}{\sigma^{2}B\delta {\left(i,j\right)}^{2}}\right)$$

Equation (Equation 10) represents Shannon’s second law, *B* represents the bandwidth of the communication channel, p[t] represents the transmit power at time *t*, $\eta_{0}$ represents the channel gain, $\sigma^{2}$ represents the noise power spectral density, and $\delta {\left(i,j\right)}^{2}$ represents the distance between nodes *i* and *j*. Shannon’s second law states that the capacity of a channel between two nodes is inversely proportional to the distance between them. Based on this, the average distance from non-backbone nodes to backbone nodes is used as a measure of network communication capability. Thus, this multi-objective optimization problem can be represented as follows:(11)$$\begin{array}{c} opt.\left\{\begin{array}{c} \min\sum_{i=1}^{N}d_{CDS}^{t}=\frac{\sum_{i=1}^{N}d_{v_{i}\to CDS}}{N}\\ \max R_{\mathrm{cov}}=\frac{\sum_{j=1}^{l_{1}\times l_{2}}C_{p}\left(Z,m_{j}\right)}{l_{1}\times l_{2}} \end{array}\right.\\ s.t.\;\lambda_{2}\left(L\right)>0 \end{array}$$

During the network topology maintenance phase, when the overall performance of the network is good, minor adjustments to the positions of network nodes under low energy consumption are sufficient. Based on this, this paper proposes optimizing the network coverage rate and node movement distance as objective functions. Similarly, through multiple rounds of iteration using the multi-objective enhanced marine predator algorithm, the positions of non-backbone nodes in the network are optimized. The problem is mathematically modeled as a constrained dual-objective optimization problem, as shown below:(12)$$\begin{array}{c} opt.\left\{\begin{array}{c} \min\sum_{v_{p}\in V}∥s_{t_{k+1}}\left(v_{p}\right)-s_{t_{k}}\left(v_{p}\right)∥\\ \max R_{\mathrm{cov}}=\frac{\sum_{j=1}^{l_{1}\times l_{2}}C_{p}\left(Z,m_{j}\right)}{l_{1}\times l_{2}} \end{array}\right.\\ s.t.\;\lambda_{2}\left(L\right)>0 \end{array}$$
where $s_{t+1}\left(v_{p}\right)$ and $s_{t_{k}}\left(v_{p}\right)$ represent the positions of node $v_{p}$ at time t+1 and $t_{k}$, respectively.

## 3. Virtual Backbone Network Based on CDS

### 3.1. Backbone Network Construction Phase

During the backbone network construction phase, this paper proposes a method that combines minimum spanning tree (MST) and local routing information to form a minimum CDS. By applying the MST algorithm, it is possible to select a set of nodes with the minimum weight as the backbone nodes. To ensure the minimal number of backbone nodes, this paper selects node degree as the key indicator for weighting selection.

From the pseudo-code of Algorithm 1, we can see that the algorithm consists of three steps.
**Algorithm 1** MST-based CDS backbone network construction algorithm1:**Input:** Network topology diagram for the current time slot G(V,E)2:**Output:** The best one in the current time slot $S_{cds}^{t}$ initialization3:**Initialization:** Calculate the node degree, edge weight, and node weight of each node in the network topology graph4:**for** x=1 to *N* **do**5:    Find $U=\arg\max\left[w^{†}\left(N_{1}^{t}\left(v\right)\right)\right]$, $\left\{v\in V_{NB},u\notin V_{NB}|u\in U\right\}$6:    **if** The number of nodes in set *U* is 1 **then**7:        $v_{q}\leftarrow v_{q}^{\prime }$8:    **else**9:        $v_{q}=\arg\max\left[w\left(N^{t}\left(v_{k}\right)\right)\right]$, $v_{k}\in U$ // Links with maximum node weight10:    **end if**11:    $V_{cds}^{x}\leftarrow u$, $E_{cds}^{x}\leftarrow <v_{p},v_{q}>$12:    **repeat**13:        **if** $D\left(v_{p}\right)=D\left(v_{q}\right)=1$ and $<v_{p},v_{q}>$ exist **then**14:            Deleting nodes $v_{p}$, $v_{q}$15:        **end if**16:    **until** $V_{cds}^{x}=V$17:**end for**

1. Network Initialization and Construction of Adjacency Matrix: (a) Node Broadcast Communication: Each node in the network communicates with other nodes by broadcasting HELLO messages to collect information about neighboring nodes. This information includes node degree, node weights, and other critical parameters essential for subsequent network topology analysis and optimization. (b) Confirmation of Bidirectional Communication: Upon receiving HELLO messages from neighboring nodes, nodes reply with maximum power to ensure reliable bidirectional communication, completing the network initialization process. (c) Acquisition of Adjacency Matrix: Through periodic message exchanges, each node eventually obtains detailed information about directly connected neighboring nodes, thus constructing the adjacency matrix that represents the current network topology. This matrix forms the basis for constructing the minimum spanning tree (MST) and minimum CDS.

2. Construction of Minimum Spanning Tree for Generating CDS: (a) Setting Edge Weights and Node Weights: After obtaining the adjacency matrix, the algorithm sets edge weights $w(e_{<v_{p},v_{q}>})$ based on the connections between nodes and their node degrees $w(v_{p})$. (b) Application of MST Algorithm: Choosing any node in the network as the starting point, the MST algorithm traverses the entire network topology. During this process, the MST algorithm prioritizes selecting edges with smaller weights to construct the tree, ensuring that the sum of edge weights in the resulting minimum spanning tree is minimized while covering all nodes. (c) Construction of CDS Set: Through the MST algorithm, a series of minimum CDSs are obtained. Each CDS is a candidate for the backbone network, ensuring network connectivity and coverage by including key nodes and connections.

3. Selection and Optimization of Backbone Network: (a) Calculation of CDS Weights: For each CDS in $S_{cds}^{t}$, calculate the sum of weights of its dominating nodes to obtain the weight of the CDS $w(CDS_{k})$. (b) Selection of Optimal CDS: Among all generated CDSs, the algorithm selects the CDS with the highest weight as the backbone network at the current time.

Through the above steps, this paper effectively constructs a minimum CDS based on minimum spanning tree and local routing information, thereby forming an efficient and stable network backbone. This approach better adapts to the complex and dynamic communication environments and requirements of tactical edge networks. The specific algorithm flow is as follows:

### 3.2. Backbone Network Maintenance Phase

During the backbone network construction phase, all potential CDSs have been identified and sorted based on their weights. In the simulation work of this paper, a series of network topology snapshots are obtained by periodically sampling the runtime of the entire network, enabling analysis of the continuously changing network structure. In practical applications, especially in dynamic battlefield environments, algorithms need to adjust the frequency of updating topology snapshots based on field conditions.

Figure 4 is a snapshot of the network topology at different times. At time $t_{k}$, the network topology snapshot is shown in Figure 4a. The dark-colored nodes 2, 4, 5, and 6 represent the virtual backbone network formed at $t_{k}$. As time progresses, at time $t_{k+1}$, the network topology changes as shown in Figure 4b. The changes include the disconnection between nodes 3 and 4, as well as between nodes 3 and 5. Additionally, a new connection is established between nodes 7 and 9. The backbone nodes at time $t_{k+1}$ are now nodes 2, 5, and 7.

An effective backbone network maintenance strategy is crucial for ensuring stable network operation. Most existing research focuses only on fault tolerance mechanisms during the network establishment phase, while overlooking issues caused by dynamic node changes during the maintenance phase. Therefore, in the context of tactical edge networks, developing an efficient update scheme for CDSs is particularly critical. Addressing this need, this paper introduces a novel multi-CDS scheduling algorithm for backbone network maintenance. This algorithm flexibly selects alternative CDSs for node status updates when the backbone network topology undergoes dynamic changes, while striving to minimize costs during the update process. This approach effectively addresses various failure scenarios that may occur in the backbone network. The specific steps of the algorithm are as follows:

1. Periodic Topology Updates: As the network continues to run, each node periodically broadcasts HELLO messages. Neighboring nodes receiving these messages respond with their state information at maximum power, thereby achieving real-time updates of network topology information. Additionally, the algorithm uses continuous topology snapshots updated at fixed intervals $t_{d}$ to ensure the accuracy and timeliness of network state.

2. Constructing MCDS Backbone Network: Firstly, the minimum spanning tree (MST) algorithm traverses all nodes in the network to gradually form the MCDS set. Then, for each generated CDS, the total weight $w(CDS_{k})$ is calculated by summing the weights of all nodes in the set. Finally, the CDS with the lowest total weight is selected as the backbone network structure.

3. Dynamic Response to Network Topology Changes: When the network topology changes, the existing backbone network may no longer fully cover the entire network. In such cases, the algorithm applies the principle of maximum similarity to select suitable CDSs from the remaining MCDSs, minimizing the number of nodes that need to be updated. This approach swiftly restores network coverage and connectivity, as illustrated in Figure 5.

4. Node Role Updating and Adjustment: Upon selecting the new CDS, the algorithm adjusts the roles of nodes accordingly based on the composition of the new set. Nodes that were originally not part of the backbone network are promoted to backbone nodes, while those that have become ineffective or are no longer in critical positions are demoted to ordinary nodes. This ensures the optimization of the backbone network structure.

5. Periodic Updates of MCDSs: To effectively respond to continuous dynamic changes in the network, the algorithm periodically checks and adjusts the alternative CDSs (MCDSs) according to a predefined schedule. Once the number of available MCDSs falls below a critical threshold of 2, the algorithm initiates the MCDS construction process again. This ensures that even under significant changes in network conditions, the backbone network maintains its coverage and connectivity. When the network topology changes, the scheduling process is re-executed based on Algorithm 2, which is the backbone network maintenance algorithm based on multi-CDS scheduling, to adapt to the new network environment.
**Algorithm 2:** Backbone network maintenance algorithm based on multi-CDS scheduling1:**Input:** MCDS collection, network topology $G(V^{t},E^{t})$, Update Cycle $t_{d}$2:**Output:** Scheduling in $t_{k}$ time slot3:    **Initialization:** $CDS_{t}\leftarrow \min\left\{W\left(MCDS\right)\right\}$4:**while** true **do**5:   **if** $v_{p}\notin V_{CDS_{t_{k}}}$ **then**6:     $S_{cds}\leftarrow S_{cds}\setminus CDS_{i}$7:     $CDS_{t_{k}}\leftarrow \arg\max\left(T_{cost}\left(S_{cds},CDS_{t_{k}}\right)\right)$8:   **end if**9:   **if** $CDS_{t_{k}}$ The current backbone network connection edge or node fails, $v_{p},v_{q}\in V_{CDS_{t_{k}}}$ **then**10:     $\left\{CDS_{k}\right\}=\arg\left\{\left(v_{p},v_{q}\right)\in E_{CDS_{k}}\right\}$11:     $S_{cds}\leftarrow S_{cds}\setminus CDS_{k}$12:     $CDS_{t_{k}}\leftarrow \arg\max\left(T_{cost}\left(S_{cds},CDS_{t_{k}}\right)\right)$13:   **end if**14:**end while**

## 4. Non-Backbone Node Optimization Strategy Based on MOEMPA

To further enhance various aspects of the current network topology in the present time slot, such as network coverage, resilience, and communication capability, it can be modeled as a constrained multi-objective optimization problem. In order to obtain an approximate optimal solution to this multi-objective optimization problem within limited computational time, this paper adopts the multi-objective enhanced marine predator algorithm [22] (MOEMPA) to optimize the positions of non-backbone nodes in the virtual backbone network. MOEMPA modifies the fitness function, population update mechanism, etc., based on EMPA, and introduces top predator selection strategies and an external archive set (Archive) to store Pareto-optimal solutions. Through multiple iterations of population optimization, MOEMPA ultimately obtains a Pareto solution set, enabling multi-objective decision-making to construct an optimal topology structure that meets resilience and communication requirements.

### 4.1. Introduction of External Archive Set

This study introduces an external archive set aimed at accumulating all explored non-dominated Pareto optimal solution sets. The maximum capacity of this archive set is fixed and typically set to half the size of the population. The archive set is responsible for periodically collecting solutions from the current population and updating the archive by comparing them with newly generated solutions. The update rules are as follows:

1. If a newly generated solution is dominated by any solution in the external archive set, it is not added to the archive.

2. If a new solution dominates one or more solutions in the archive set, the inferior solutions are removed from the archive, replaced by the new solution.

3. If a new solution is not dominated by any solution in the archive, it is included in the archive.

According to the above update rules, each solution in the archive set consistently dominates all other solutions in the multi-objective optimization problem. However, when the archive capacity reaches its limit and a new solution meets the update criteria, without action, excellent new solutions may be overlooked. Therefore, one solution strategy is to randomly remove a solution from the archive, minimizing the impact on the distribution of the solution set. The solution selected for removal should minimally affect the distribution of the solution set. To evaluate the distribution of solutions in the archive set, this paper uses the crowding distance $\vec{d}$, calculated as the number of neighboring solutions within this range:(13)$$\vec{d}=\frac{\vec{\max}-\vec{\min}}{Archive\_size}$$
where $\vec{\max}$ and $\vec{\min}$ are vectors storing the maximum and minimum values of each solution, and Archive_size is the size of the archive.

Figure 6 illustrates the process of removing solutions from the archive set. In the context of minimizing objectives, the small circles depicted in the figure represent individual solutions within the archive. Each solution’s crowdedness in its respective area is evaluated by calculating the number of neighboring solutions within a certain distance range (represented by the large circle) around each solution. In areas with high congestion, it means that there are more solutions in that region, making these solutions more likely to be selected for removal. Conversely, if the number of neighboring solutions is smaller, it indicates that the region where the solution is located is relatively sparse, and such solutions should be preserved. In the figure, the red circles represent the current non-dominated solutions in the archive, while the blue circles represent newly generated solutions or neighboring solutions that need to be evaluated.

Next, this study employs the number of neighboring solutions as a quantitative measure to assess the crowdedness level of each solution in the solution set. Based on the obtained crowdedness, all solutions are ranked, and a roulette wheel selection mechanism determines the solutions to be eliminated. The probability of each solution being selected is given by the following:(14)$$P_{i}=\frac{N_{i}}{C}$$
where *C* is a constant greater than 1, and $N_{i}$ represents the number of neighboring solutions for the $P_{i}$ solution in the archive. $P_{i}$ denotes the probability of selecting the *i* solution. Equation (Equation 14) implies that the higher the crowdedness of a solution, the greater its probability of being selected for removal from the archive. Ultimately, the selected solution is removed from the archive, while any new solution meeting the update criteria is added to the archive.

### 4.2. Top Predator Selection Strategy

In the multi-objective marine predator algorithm, the top predator represents the most experienced individual in the population, playing a pivotal role in leading other individuals to find food, significantly influencing the convergence of the algorithm. In single-objective optimization, individual fitness values can be directly compared, and the individual with the highest fitness is selected as the top predator. However, in the context of multi-objective optimization, due to the diversity of objective functions, solutions cannot be directly compared. In such cases, the Pareto dominance concept is used to rank solutions. As previously described, the archive preserves all non-dominated solutions found so far, and the top predators are also among these non-dominated solutions. Therefore, the top predator can be selected from the archive.
(15)$$f_{i}=N_{i}$$

From Equation (Equation 15), it is evident that the updated fitness of each solution is determined by the number of neighboring solutions. Applying this formula allows for the computation of a new fitness value for each solution and selection of the one with the lowest fitness as the top predator. Specifically, in the multi-objective enhanced marine predator algorithm (MOEMPA), more than one top predator may be selected. The elite selection mechanism tends to favor solutions in sparsely populated regions of the archive, which is highly advantageous for improving the algorithm’s performance and optimizing the distribution of solutions. After selecting the top predators, the construction of the Elite matrix in MOEMPA is as follows:(16)$$\;Elite\;={\left[\begin{array}{cccc} X_{1,1}^{1} & X_{1,2}^{1} & \ldots & X_{1,D}^{1}\\ \ldots & \ldots & \ldots & X_{2,D}^{2}\\ \ldots & \ldots & \ldots & \ldots \\ X_{k,1}^{k} & X_{k,2}^{k} & \ldots & X_{k,D}^{k}\\ X_{k+1,1}^{k} & X_{k+1,2}^{k} & \ldots & X_{k+1,D}^{k}\\ \ldots & \ldots & \ldots & \ldots \\ X_{N,1}^{m} & X_{N,2}^{m} & \ldots & X_{N,D}^{m} \end{array}\right]}_{N\times D}$$
where $X^{1},X^{2},\ldots ,X^{m},\ldots ,X^{k}$ represent the top predators selected using the top predator selection strategy, where 1<m<k,k≤Archive_size. The matrix provides the current best prey locations for each predator.

### 4.3. Algorithm Steps

Based on the MOEMPA-based optimization algorithm for non-backbone node positioning in tactical edge networks, here are the specific steps:

1. Parameter Setting: Input parameters include positions of non-backbone nodes, monitoring area range, sensing radius, population size, number of solutions in the external archive, and total number of iterations.

2. Objective Function Definition: Define the fitness function based on coverage rate in the monitoring area of tactical edge networks. Initialize positions of randomly scattered nodes and calculate initial area coverage based on a joint perception model. Set fitness functions for different fault levels (small and large scales) as optimization objectives using Equations (Equation 11) and (Equation 12).

3. Update External Archive: Update the archive with Pareto-optimal solutions. If the archive exceeds its capacity, use a roulette wheel selection mechanism to remove solutions and check if any individuals in the population are out of bounds.

4. Determine Initial Prey Matrix and Elite Predator Matrix: Calculate fitness values of the prey matrix, record the best positions, and compute the elite predator matrix.

5. Update Prey Matrix: Before the algorithm starts, the population is initialized, and different updating methods are applied during various iteration periods. In the early iterations, prey undergo Brownian motion for global search. In the mid-iterations, prey use Lévy flight, while predators employ Brownian motion. In the late iterations, predators switch to Lévy flight strategy for local search. This approach aims to address issues related to eddy formation and fish aggregating device (FAD) effects, which in the context of intelligent optimization algorithms refers to a type of vortex effect.

6. Proceed to Next Iteration: Increment the iteration counter: Iter=Iter+1.

7. Output Current Pareto Front: Output the current Pareto front and select the optimal deployment plan for non-backbone nodes.

## 5. Simulation Analysis

This chapter focuses on simulation analysis of dynamic topology optimization in tactical edge networks, particularly examining network failures caused by changes in network topology strength. Simulations are conducted with a network coverage threshold set at 80%, testing both large-scale and small-scale fault scenarios.

### 5.1. Experimental Setup

To better simulate the fault recovery capabilities of tactical edge networks in different scenarios, this study sets a standard tactical edge network as the baseline model. Teams, consisting of 20 homogeneous nodes each with a sensing radius of 100m, are tested in network fault scenarios across three area sizes: 800 m × 800 m, 1000 m × 1000 m, and 1200 m × 1200 m. The testing involves 1, 2, and 3 teams, respectively, as detailed in Table 1.

The relevant parameter settings of the MOEMPA algorithm are shown in Table 2. In our study, we selected 200 iterations for the MOEMPA algorithm based on several considerations. Firstly, the choice of 200 iterations balances computational cost and precision. This number allows the MOEMPA algorithm to converge efficiently to a solution while maintaining a manageable computational burden, ensuring that the results are both accurate and cost-effective. Secondly, the decision is supported by empirical evidence from previous experiments and studies. It has been demonstrated that 200 iterations are sufficient to achieve stable and high-quality results across various test scenarios, making it a well-established choice. Finally, our experiments show that 200 iterations are typically adequate for the algorithm to reach convergence. Beyond this number, additional iterations do not significantly enhance the solution quality. Furthermore, limiting the iterations to 200 helps avoid overfitting to specific problem instances or training data, thereby enhancing the algorithm’s ability to generalize to new and unseen scenarios.

Overall, the selection of 200 iterations for the MOEMPA algorithm is a well-founded decision that optimizes performance and efficiency based on both theoretical and practical considerations.

### 5.2. Large-Scale Failure Testing

If the current network coverage falls below 80%, indicating a need for topological reconstruction, the process begins with selecting backbone nodes using the CDS algorithm to establish a virtual backbone network. Subsequently, optimizing the positions of non-backbone nodes is carried out using the multi-objective enhanced marine predator algorithm (MOEMPA). The optimization aims to minimize the average distance from non-backbone nodes to backbone nodes (avg-d), with current network algebraic connectivity as a constraint. This iterative process aims to rapidly restore network functionality, enhancing both network communication and coverage capabilities. The average network coverage is represented by avg-cov.

(1) Scenario 1: 800 m × 800 m Monitoring Area

Specific parameters for this scenario include monitoring area size of 800 m × 800 m, 20 nodes, and a sensing radius of 100 m.

Figure 7a,d depict the initial coverage area map and node connectivity graph for this scenario, showing a scattered distribution of network nodes and poor network connectivity. Significant coverage holes and redundancy are observed in the central part of the monitoring area, leading to a relatively low overall network coverage rate. Figure 7b illustrates the use of the CDS algorithm to select 10 backbone nodes (highlighted in red) from an initial set of 20 nodes to establish an efficient and effective backbone network.

According to Table 3 and Figure 7c,e, after 200 rounds of MOEMPA iteration optimization for non-backbone nodes, significant reductions in coverage redundancy and holes are observed in the original monitoring area. The coverage rate notably improves from an initial 56.11% to 88.52%, marking an increase of nearly 30%, which is significantly higher than the initial state. Furthermore, the average distance from non-backbone nodes to backbone nodes decreases from 405.44 m to 383.33 m, indicating a marked improvement in overall node connectivity compared to before optimization.

(2) Scenario 2: 1000 m × 1000 m Monitoring Area

Specific parameters for this scenario are as follows: monitoring area size of 1000 m × 1000 m, 40 nodes, and a sensing radius of 100 m.

According to Table 4 and Figure 8, it can be observed that in the event of a large-scale network failure with an initial coverage rate of 72.64%, employing the CDS algorithm selected 21 backbone nodes from an initial distribution of 40 nodes to establish the backbone network. Following 200 rounds of MOEMPA algorithm optimization, non-CDS nodes significantly reduced coverage redundancy and coverage holes at the bottom and right sides of the monitoring area, increasing the coverage rate to 88.52%, which is significantly higher than the initial state. Additionally, the average distance from non-CDS nodes to CDS nodes decreased from 556.05 m to 491.76 m, a reduction of approximately 10%, thereby enhancing the network’s resilience and communication capabilities to a certain extent.

(3) Scenario Three: 1200 m × 1200 m monitoring area

Specific parameters for this scenario are as follows: monitoring area size of 1200 m × 1200 m, 60 nodes, and a sensing radius of 100 m.

Figure 9a,d depict the initial coverage area map and node connectivity graph for this scenario, showing a scattered distribution of network nodes and poor network connectivity. Significant coverage holes and redundancy are observed in the central part of the monitoring area, resulting in a relatively low overall network coverage rate. Figure 9b shows the use of the CDS algorithm to select 21 backbone nodes (highlighted in red) from an initial distribution of 60 nodes to establish an efficient and effective backbone network.

According to Table 5 and Figure 9c,e, after 200 rounds of MOEMPA iteration optimization for non-backbone nodes, significant reductions in coverage redundancy and holes are observed in the original monitoring area. The network coverage rate improves from an initial 72.92% to 86.28%. Furthermore, the average distance from non-backbone nodes to backbone nodes decreases from 670.01 m to 615.00 m, indicating a noticeable improvement in overall node connectivity compared to before optimization. This enhancement significantly boosts the network’s resilience and communication capabilities.

### 5.3. Small-Scale Failure Testing

If the current network coverage rate is determined to be above 80%, the network enters a topology maintenance phase. Firstly, using a multi-CDS scheduling algorithm based on the maximum similarity principle, the backbone network topology is updated for the current time slot. Subsequently, aiming to optimize the current network coverage rate and node mobility distance with current network algebraic connectivity as a constraint, the positions of non-backbone nodes are fine-tuned using MOEMPA. This aims to further enhance the network coverage rate while minimizing energy consumption, thereby constructing the optimal topology structure for the current time slot.

Figure 10a–c represent the initial topology under Scenario 1, the topology after the CDS backbone network update, and the topology optimized by MOEMPA, respectively. Figure 10d–f show the corresponding topology under Scenario 2, and Figure 10g–i show the corresponding topology under Scenario 3.

After optimizing the positions of newly added non-backbone nodes using the MOEMPA algorithm, it was observed that node distribution became more uniform across three different scenarios, with significant reductions in coverage holes and redundancy. According to the data in Table 6, it is evident that in these three scenarios, the network coverage rate increased from an initial 80.17% to 92.28%, from 82.52% to 91.23%, and again from 82.52% to 91.23%, achieving a significant improvement of approximately 10% in each case compared to before optimization. Additionally, the average node movement distance remained at a low level, further enhancing both the coverage efficiency and network resilience.

Further examination of the results shown in Figure 10 reveals that after updating the virtual backbone network, the initial coverage areas became more rational, and the layout of the backbone network was optimized. Moreover, leveraging the maximum similarity mechanism minimized the cost of updating the backbone network. In these three different scenarios, after the backbone network update, 3, 4, and 11 original backbone nodes were reassigned as non-backbone nodes, while 2, 2, and 6 original non-backbone nodes were upgraded to backbone nodes. The results indicate a reduction in the total number of backbone nodes of 1, 2, and 5, respectively, demonstrating effective reduction of network update costs and resource expenditures while ensuring network connectivity.

Furthermore, these changes not only highlight the effectiveness of the multi-CDS scheduling algorithm in updating the backbone network but also showcase the strong capability of the MOEMPA algorithm in maintaining network stability and adaptability. Following the backbone network update, by dynamically optimizing and adjusting the distribution of nodes in the network, the MOEMPA algorithm flexibly adjusts based on actual network conditions, ensuring optimal network operation in the face of various dynamic changes. This capability is particularly crucial for tactical edge networks that require stable operation in complex and rapidly changing environments. Therefore, the MOEMPA algorithm not only improves network coverage and resilience but also provides robust support for long-term stable operation and resource optimization management of the network.

### 5.4. Comparative Algorithm Testing

To further validate the effectiveness of the MOEMPA algorithm for optimizing node positions, experiments were conducted using different multi-objective optimization algorithms: multi-objective particle swarm optimization (MOPSO) [23], multi-objective whale optimization algorithm (MOWOA) [24], and multi-objective gray wolf optimizer (MOGWO) [25]. These algorithms were compared and analyzed for node movement problems in the experimental module. The parameters for each algorithm are detailed in Table 7.

Similarly, these multi-objective algorithms were applied to test Scenario 2 (1000 m × 1000 m monitoring area) under network large-scale failure conditions. After multiple rounds of experiments, the optimization results of each algorithm are presented in Table 8. From Table 8, it can be observed that the node movement strategy based on the MOEMPA algorithm yielded the best optimization results in terms of both average coverage rate and average distance from non-CDS nodes to CDS nodes. Specifically, compared to MOPSO, MOGWO, and MOWOA, MOEMPA improved the coverage rate by 14.50%, 11.54%, and 9.57%, respectively, and reduced the average distance to CDS nodes by 28.31 m, 57.59 m, and 8.73 m, respectively.

To comprehensively evaluate the performance and specific trends of the MOEMPA, MOGWO, MOWOA, and MOPSO algorithms during the iterative process, this study compared their convergence curves in terms of network coverage rate and average node movement distance. Figure 11a illustrates that the convergence curves of average coverage rate for all four algorithms exhibit initial rapid improvement, followed by a gradual slowdown and eventual stabilization. This reflects the algorithms’ strong global search capability in the early stages, enabling them to quickly approach optimal solutions, while demonstrating robust local search capability in later stages to achieve convergence. Importantly, the MOEMPA algorithm proposed in this study outperforms the other three algorithms in terms of performance. This advantage stems from the top predator selection strategy integrated into MOEMPA, which enhances convergence speed and solution distribution quality, making the algorithm more effective in pinpointing optimal solutions.

Furthermore, Figure 11b displays the variation curves of average node movement distance for the four algorithms during the optimization process. The results indicate that throughout the entire iteration process, the MOEMPA algorithm consistently achieves the best optimization in both network coverage rate and average node movement distance, thereby validating the superiority of this algorithm across multiple aspects.

## 6. Conclusions

This study proposes a dynamic network topology optimization method based on a virtual backbone network for tactical edge networks, addressing issues of network vulnerability and low communication efficiency due to frequent node mobility during operational phases. The method is divided into two stages: topology reconstruction and topology maintenance. During the reconstruction stage, backbone nodes are selected using CDS algorithms to establish a virtual backbone network within the designated monitoring area. Subsequently, a multi-objective enhanced marine predator algorithm iteratively optimizes the positions of non-backbone nodes. The optimization aims to maximize current network coverage and minimize the average distance from non-backbone nodes to backbone nodes, while adhering to current network connectivity constraints. In the maintenance stage, a multi-CDS scheduling algorithm updates the backbone network topology based on the principle of maximum similarity. Further optimization involves fine-tuning the positions of non-backbone nodes using a multi-objective enhanced marine predator algorithm, focusing on optimizing current network coverage and node movement distances under connectivity constraints. Experimental results validate the effectiveness of the proposed method in addressing both large-scale and small-scale network failure scenarios. The approach enables rapid interconnection, enhances network coverage, and constructs an optimal network topology.

## Figures and Tables

**Figure 1 sensors-24-05489-f001:**
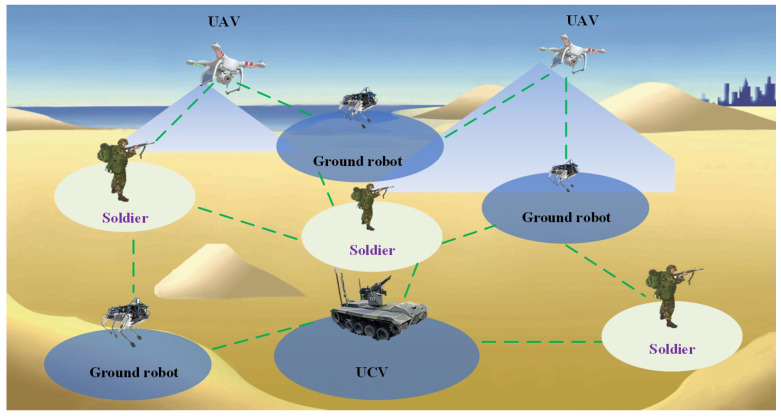
Structure diagram of tactical edge network.

**Figure 2 sensors-24-05489-f002:**
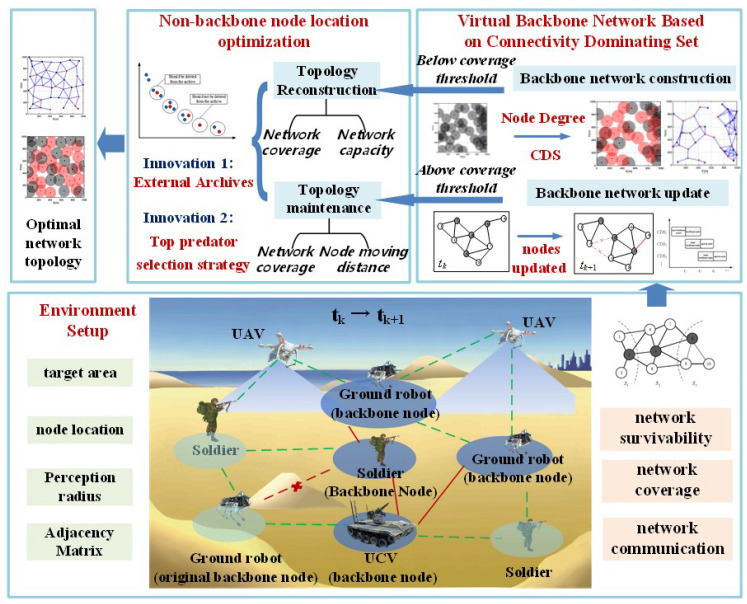
Dynamic topology optimization solution for tactical edge networks.

**Figure 3 sensors-24-05489-f003:**
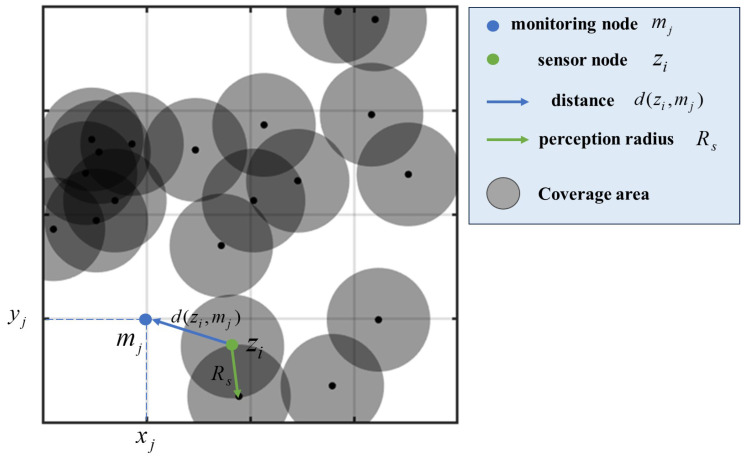
Schematic diagram of network coverage model.

**Figure 4 sensors-24-05489-f004:**
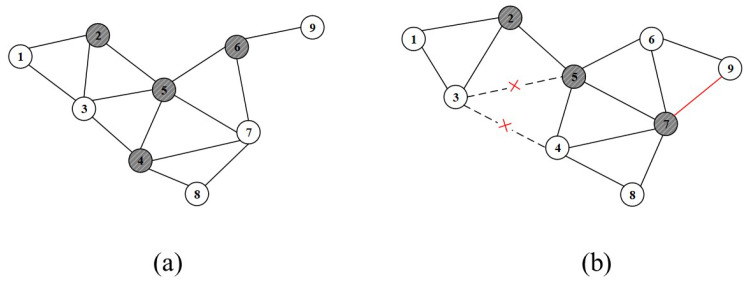
Network topology snapshots at (**a**) time $t_{k}$, (**b**) time $t_{k+1}$.

**Figure 5 sensors-24-05489-f005:**
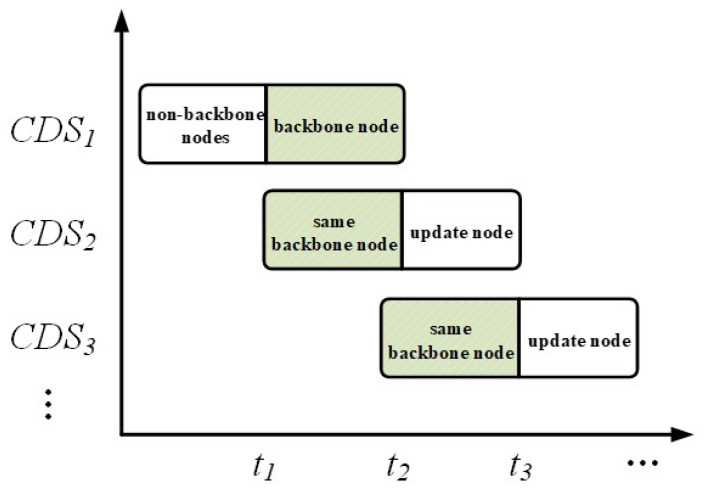
Maximum similarity principle update mechanism.

**Figure 6 sensors-24-05489-f006:**
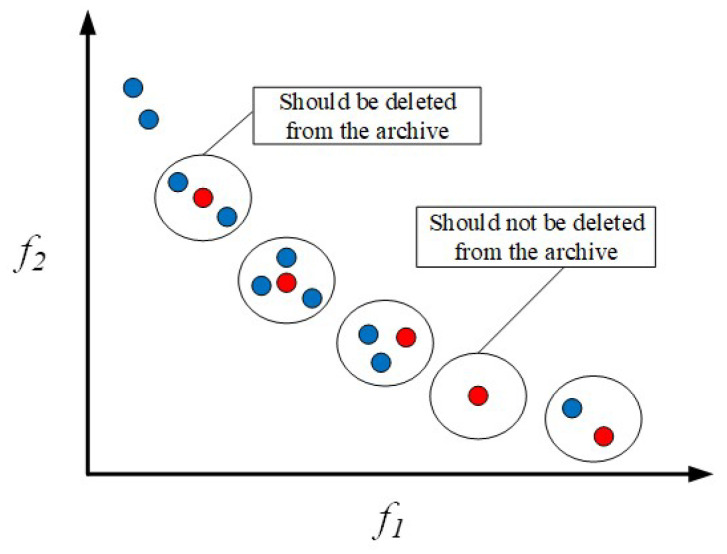
Solution removal mechanism when the archive set is full.

**Figure 7 sensors-24-05489-f007:**
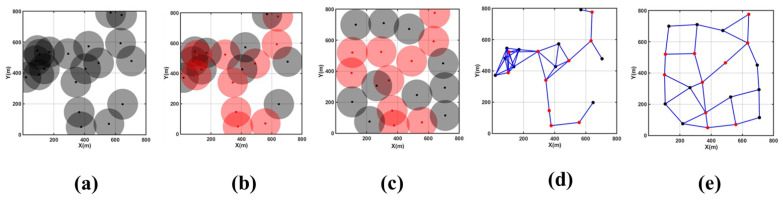
Coverage area optimization diagram for large-scale failure Scenario 1. (**a**) Initial coverage area, (**b**) CDS backbone network structure, (**c**) MOEMPA location optimization, (**d**) initial node connectivity graph, (**e**) optimized node connectivity graph.

**Figure 8 sensors-24-05489-f008:**
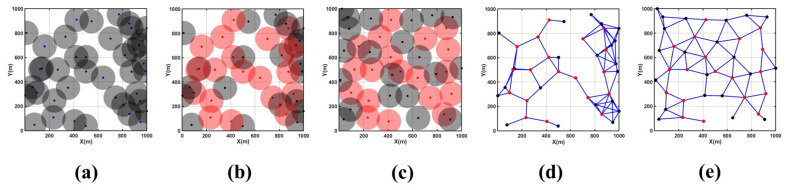
Coverage area optimization diagram for large-scale failure Scenario 2. (**a**) Initial coverage area, (**b**) CDS backbone network structure, (**c**) MOEMPA location optimization, (**d**) initial node connectivity graph, (**e**) optimized node connectivity graph.

**Figure 9 sensors-24-05489-f009:**
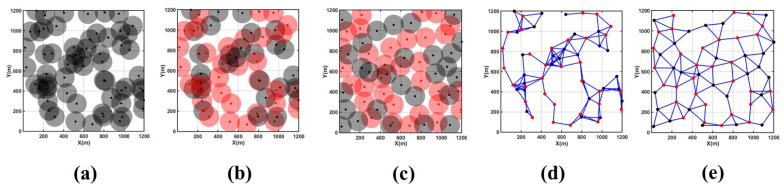
Coverage area optimization diagram for large-scale failure Scenario 3. (**a**) Initial coverage area, (**b**) CDS backbone network structure, (**c**) MOEMPA location optimization, (**d**) initial node connectivity graph, (**e**) optimized node connectivity graph.

**Figure 10 sensors-24-05489-f010:**
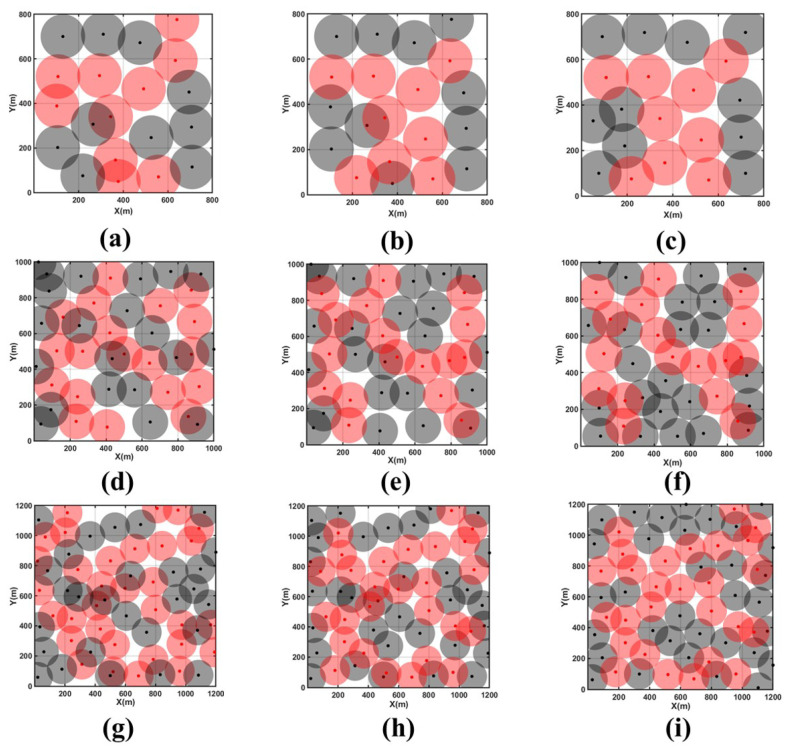
Coverage optimization in three scenarios. (**a**) Initial topology in Scenario 1, (**b**) CDS backbone network update in Scenario 1, (**c**) MOEMPA location optimization in Scenario 1, (**d**) initial topology in Scenario 2, (**e**) CDS backbone network update in Scenario 2, (**f**) MOEMPA location optimization in Scenario 2, (**g**) initial topology in Scenario 3, (**h**) CDS backbone network update in Scenario 3, (**i**) MOEMPA location optimization in Scenario 3.

**Figure 11 sensors-24-05489-f011:**
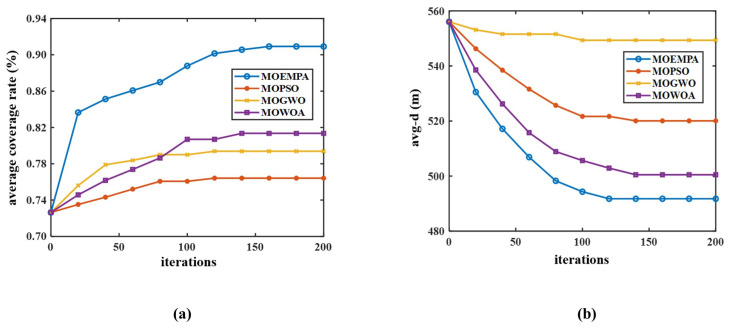
Comparison of optimization results of various multi-objective algorithms. (**a**) Comparison of coverage and iteration number, (**b**) node distance–iteration number comparison chart.

**Table 1 sensors-24-05489-t001:** Scenario settings.

Number of Teams	Area Size (m^2^)	Number of Nodes	Perception Radius (m)
1	800 × 800	20	100
2	1000 × 1000	40	100
3	1200 × 1200	60	100

**Table 2 sensors-24-05489-t002:** MOEMPA algorithm parameter settings.

Parameter	Numeric
Population size	30
File size	20
Iteration number	200
Probability of eddy current influence FADs	0.2
Constant P	0.5

**Table 3 sensors-24-05489-t003:** Comparison of scene 1 before and after optimization.

Target Name	Unoptimized	Optimized
avg-cov	56.11%	82.23%
avg-d	405.44 m	383.33 m

**Table 4 sensors-24-05489-t004:** Comparison of scene 2 before and after optimization.

Target Name	Unoptimized	Optimized
avg-cov	72.64%	88.52%
avg-d	556.05 m	491.76 m

**Table 5 sensors-24-05489-t005:** Comparison of scene 3 before and after optimization.

Target Name	Unoptimized	Optimized
avg-cov	72.92%	86.28%
avg-d	670.01 m	615.00 m

**Table 6 sensors-24-05489-t006:** Small-scale fault optimization results.

Scene Type	Avg-d	Unoptimized avg-cov	Optimized avg-cov
scene 1	68.24 m	82.23%	89.78%
scene 2	104.86 m	88.52%	93.42%
scene 3	112.81 m	86.28%	94.12%

**Table 7 sensors-24-05489-t007:** Multi-objective algorithm parameter setting.

Parameter	Numeric
Iterations	200
Population size	30
External archive size	30

**Table 8 sensors-24-05489-t008:** Comparison of algorithm optimization effects.

Algorithm	Avg-cov	Avg-d
Unoptimized	72.64%	556.05 m
MOEMPA	90.92%	491.76 m
MOPSO	76.42%	520.07 m
MOGWO	79.38%	549.35 m
MOWOA	81.35%	500.49 m

## Data Availability

All relevant data are within the manuscript.

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
