# Peer review of "A Dynamic Topology Optimization Method for Tactical Edge Networks Based on Virtual Backbone Networks"

_sensors, 2024, doi:10.3390/s24175489_

Round 1

Reviewer 1 Report

Comments and Suggestions for Authors

This article is devoted to a topic that is very relevant in our time. The work presents and analyzes in detail new results significant for the development of science and technology.

First, the existing approaches to the organization of communication on the battlefield were analyzed in detail, the gaps that need to be solved were clearly defined. Structure Diagram of Tactical Edge Network was analyzed. This network system deeply integrates the technological characteristics of Wireless Sensor Networks (WSN) and Mobile Ad Hoc Networks (MANET). During the operation of the tactical edge network, nodes are susceptible to enemy attacks, leading to frequent node mobility and resulting in changes in the network's center of gravity. In such a highly dynamic and topologically variable battlefield environment, the network may face issues like node failures, causing a decline in network performance or even large-scale breakdowns.

There are some comments/recommendations to improve the quality of the article:

1. Have added indents between text and brackets.

2. It is necessary to review the presence of punctuation marks in the formulas.

3. "where" (167) must start at the beginning of the line.

4. (186) "From the above form..." should be from the paragraph.

Comments on the Quality of English Language

There are no comments

Reviewer 2 Report

Comments and Suggestions for Authors

The work submitted is very well explained and detailed, but some small errors should be corrected:

- There are missing spaces between words:

- Page 1 line 6 “Set (CDS)”;

- Page 3, line 81 “[17] .”, line 83 “performance. FU D”, line 90 “[20] .”, line 98 “optimization. Based”;

- Page 13, line 422 “nodes (avg-d)”, line 429 “(a), (b)”

- On page 2, line 44, “CDS” should be defined, even though it already is in the abstract.

- Page 3, line 79, defines “EECDS”, “DGB” (line 82), “MCDS” and “WSNs” (line 95).

- On page 3, on line 99 when it refers to Figure 1, it should be Figure 1 or Figure  2? Figure 2 should be changed and there should be an explanation in the text to make it easier to understand.

- In equation 1 (page 5) the use of “Nbr” should mean neighbor. This information should be in the text.

- Subsection 2.2 could have an image of everything that is explained in the text for a better idea of what is being covered.

- Equation 10 (page 6) presents several variables that are not explained/detailed in the text.

- Figure 3 deserved a little explanation in the text to reinforce the perception of lost links and new links.

- On page 9, line 299 should be followed by the algorithm. In order not to look out of place, I recommend referencing the algorithm and not using “as follows”.

- Page 9, line 400, defines “FAD”.

- The article would be more complete with a justification of why 200 iterations were used as a starting point.

- I think there is now a swap in the identification of tables and images:

                  - Page 14, line 435, it shouldn't be Table 3?

- Page 14, line 445, it shouldn't be Table 4?

- Page 15, line 463, it shouldn't be Table 5?

- Page 16, line 486, it shouldn't be Table 6?

- Page 16, line 491, it shouldn't be Figure 9 (c), (f) and (i)?

- Page 17, line 521, it shouldn't be Table 8?

Comments on the Quality of English Language

Already mentioned in the previous comment.
